# Developmental Evaluation of Infants Who Have Received Tadalafil in Utero for Fetal Growth Restriction

**DOI:** 10.3390/jcm9051448

**Published:** 2020-05-13

**Authors:** Shintaro Maki, Ineko Kato, Naosuke Enomoto, Sho Takakura, Masafumi Nii, Kayo Tanaka, Hiroaki Tanaka, Shinsuke Hori, Kana Matsuda, Yukito Ueda, Hirofumi Sawada, Masahiro Hirayama, Akihiro Sudo, Tomoaki Ikeda

**Affiliations:** 1Department of Obstetrics and Gynecology, Mie University Graduate School of Medicine, 2-174 Edobashi, Tsu, Mie 5148507, Japan; ikato@clin.medic.mie-u.ac.jp (I.K.); nao-e@clin.medic.mie-u.ac.jp (N.E.); s-takakura@clin.medic.mie-u.ac.jp (S.T.); m-nii1984@clin.medic.mie-u.ac.jp (M.N.); tanaka-ky@clin.medic.mie-u.ac.jp (K.T.); h_tanaka@med.miyazaki-u.ac.jp (H.T.); t-ikeda@clin.medic.mie-u.ac.jp (T.I.); 2Department of Rehabilitation, Mie University Graduate School of Medicine, 2-174 Edobashi, Tsu, Mie 5148507, Japan; s-hori@clin.medic.mie-u.ac.jp (S.H.); m-kana@clin.medic.mie-u.ac.jp (K.M.); u-yukito@clin.medic.mie-u.ac.jp (Y.U.); a-sudou@clin.medic.mie-u.ac.jp (A.S.); 3Department of Pediatrics, Mie University Graduate School of Medicine, 2-174 Edobashi, Tsu, Mie 5148507, Japan; hirosawada5@gmail.com (H.S.); hirayama@clin.medic.mie-u.ac.jp (M.H.)

**Keywords:** FGR, PDE5 inhibitor, tadalafil, KSPD

## Abstract

To assess the long-term effects of tadalafil, a therapeutic agent for fetal growth restriction (FGR), we evaluated the developmental progress of 1.5-year-old infants whose mothers had taken tadalafil during pregnancy. Twenty-four infants were assessed. We evaluated infant body weight, height, and head circumference, and performed the Kyoto Scale of Psychological Development (KSPD) test, a standardized developmental assessment covering Postural–Motor (P–M), Cognitive–Adaptive (C–A), and Language-Social (L–S) functions. The sum score was converted to a developmental quotient (DQ). The mean gestational week of the included cases was 36.1 (29–39) weeks, and the mean birth weight was 1841 (874–2646) g. Twenty-one and 20 out of the 24 cases, respectively, attained body weight and height similar to those of age-matched normal infants (within the 3rd percentile); all cases caught up in head circumference. KSPD was performed for 18 cases at 1.5 years of corrected age. The mean DQ scores were 87 (in total): 82 in P–M, 90 in C–A, and 88 in L–S. The total DQ score in one case (5.6%) was less than 70, and ranged from 70 to 85 in five cases (27.7%), and was more than 85 in 11 cases (61.1%). The growth and development of infants born of tadalafil-treated mothers seem to show good progress at a corrected age of 1.5 years.

## 1. Introduction

Fetal growth restriction (FGR) is an important perinatal disorder associated with increased perinatal mortality rates, morbidity, and neurological sequelae in infants. Previously, no established therapy existed for FGR, and it was managed by determining the optimal delivery time for the fetus.

Tadalafil, a phosphodiesterase 5 (PDE5) inhibitor, has been reported to be a potential drug for the treatment of FGR. The results of a phase II trial (TADAFER II) using tadalafil for the treatment of fetuses with early-onset growth restriction have already been published [1]. Although the Japan Agency for Medical Research and Development (AMED) recommended suspension of TADAFER II based on the results of the STRIDER-UK trial showing negative effects of sildenafil in the treatment of FGR [2] and the recruitment of new candidates for the trial was ceased, the study demonstrated a decrease in the mortality rate of FGR fetuses, neonates, and infants. In addition, contrary to the occurrence of neonatal persistent pulmonary hypertension due to sildenafil in the Dutch STRIDER study [3], no adverse events related to tadalafil were seen in mothers and neonates in the TADAFER II study. The cases were registered at less than 34 gestational weeks. Limiting the cases registered to less than 31 + 6 weeks gestational age (GA), the GA of the tadalafil treatment group was prolonged when compared to that of the conventional treatment group. Such trials report the short-term effects of tadalafil on FGR fetuses, neonates, and infants; however, the potential long-term effects of the treatment on FGR infants have not been established.

We continued to evaluate the efficacy of tadalafil, since 2015, through a retrospective study [4] and a phase I trial [5] performed at Mie University, Japan. These two studies explored the safety of tadalafil use, in addition to fetal and neonate outcomes, such as increased birth weight and fetal growth velocity; however, to date, no such evaluations have been performed for infants. Long-term neurological evaluations of infants born to mothers who took tadalafil for FGR were conducted using the Kyoto Scale of Psychological Development (KSPD). KSPD is a standardized developmental assessment scale for Japanese infants. 

The aim of the present study was to evaluate the developmental outcomes in infants whose mothers had received tadalafil for FGR treatment using KSPD at a corrected age of 1.5 years.

## 2. Experimental Section

We conducted a retrospective cohort study on live-born infants whose mothers had received 10, 20, or 40 mg tadalafil for the treatment of FGR at the Mie University Hospital between July 2015 and August 2016. All the children were scheduled for follow-ups at the Mie University Hospital. Our cohort included cases registered in our previous retrospective study [4] and in the phase I study [5]. Additional cases in which tadalafil had been administered upon the request of patients and had not been included in the two previous studies were also included in the present study. FGR was defined as an estimated fetal body weight (EFBW) below 1.5 standard deviations of the mean EFBW for the GA (according to the Japanese standard curve). Bodyweight, height, and head circumference were evaluated. Infants were considered to have caught up with age-matched normal infants when the values were above the 3rd percentile, which is thought to lead to good neurological development [6,7,8]. The KSPD was performed at 1.5 years of age (infant’s corrected age). The corrected age was adopted for the developmental evaluation of infants, considering that many deliveries in the population included in this study were preterm. It enhances the reliability of evaluations. The corrected age was adopted for the developmental evaluation of infants, considering that many deliveries in the population included in this study were preterm. It enhances the reliability of evaluations. The corrected age was calculated by subtracting the number of months of prematurity from the chronological (actual) age.

The latest version of the KSPD, standardized for 2677 Japanese children/adults, was published in 2002 [9]. The KSPD is an individualized face-to-face test performed by experienced psychologists to assess a child’s development in the following three areas: Postural–Motor functions (P–M; fine and gross motor functions); Cognitive–Adaptive functions (C–A; non-verbal reasoning or visuospatial perceptions assessed using materials); and Language–Social functions (L–S; interpersonal relationships, socializations, and verbal abilities). The test usually takes approximately 20–40 min to complete. In each of the three areas, a sum score is converted to a developmental age (DA), and an overall DA is also obtained. The DAs for three specific areas and the overall DA were then divided by the child’s chronological age and multiplied by 100 to yield four developmental quotients (DQ). Infants were assessed as normal (DQ ≥ 85), borderline (70 < DQ < 85), or abnormal (DQ ≤ 70), when compared with ordinary children of the same age. Experienced physicians and a speech pathologist performed the behavioral assessments simultaneously with the neurological examination at the Department of Pediatrics or the Department of Rehabilitation at Mie University Hospital.

Pearson’s product-moment correlation coefficient was used to evaluate linear correlations between two variables. In all analyses, *p* < 0.05 was considered to indicate significance. However, a significant linear correlation between two variables was defined as that satisfying both *p* < 0.05 and correlation coefficient (*r* value) > 0.2 or < −0.2 conditions. Analyses were performed using JMP® 14.2.0 (SAS Institute Inc., Cary, NC, USA).

This study was approved by the Ethics Committee of Mie University Hospital on April 10, 2017 (approval number: 3119) and is registered at the University Hospital Medical Information Network (UMIN, registry number: 000028773). The protocol was also approved by the institutional review boards of all the participating institutions.

## 3. Results

Figure 1 and Table 1 show the study profile and the number of tadalafil-administered FGR cases, and the actual number participating in the developmental evaluation. Thirty mothers received tadalafil from July 2015 to August 2016 at Mie University Hospital. Eleven cases were subjects in the retrospective study, 12 cases were in the phase I trial, while 7 of these mothers, who were administered tadalafil upon their request, were also included in the present study (Figure 1). A dose of 10 mg tadalafil was administered to three cases, 20 mg tadalafil to 20 cases, and 40 mg tadalafil to seven cases. One case of intrauterine fetal death occurred in the 20- and 40-mg dose groups. One case of trisomy 21 occurred in the 20-mg tadalafil group and was excluded from the present study. Two cases in the 20-mg tadalafil group and one in the 40-mg tadalafil group withdrew from the follow-up study. In total, 24 cases were evaluated at an infant corrected age of 1.5 years. KSPD was not performed for six cases at the discretion of the parents. KSPD tests were performed for a total of 18 cases. Three patients withdrew and six cases without KSPD exhibited good prognosis while visiting the hospital.

The characteristics of the cases are listed in Table 2. The mean gestational age at birth was 36.1 weeks. The mean birth weight was 1841.0 g (SD, ±2.1). The mean gestational age at the start of treatment was 30.0 weeks, and the mean estimated fetal weight was 1076.2 g (SD, ±2.0). 

The duration of treatment for each of the patients and the correlation between the GA at the start of treatment and total drug dose are illustrated in Figure 2a,b. The mean treatment period was 43.6 days. There was a significant correlation between the total dose of tadalafil and the start of treatment (*r* = −0.495, *p* = 0.009).

Table 3 and Table 4 show the numbers and rates of the caught-up cases within the 3rd percentile on body weight (BW), height, and head circumference. Twenty-one out of 24 cases (87.5%) caught up on body weight to within the 3rd percentile after a mean of 5.8 months; 20 out of 24 cases (83.3%) caught up on height after a mean of 3.9 months; and all cases caught up on head circumference after a mean of 3.8 months.

Table 5, Table 6 and Table 7 show the results of KSPD at an infant corrected age of 1.5 years. Neurodevelopmental assessment based on the KSPD test was conducted on a total of 18 cases. A DQ score of more than 85 was achieved in nine cases (50.0%) in the P–M area, 13 cases (72.2%) in the C–A area, 13 cases (72.2%) in the L–S area, and 11 cases (61.1%) in the total area (Table 5). In the 20-mg dose group, a DQ score more than 85 was achieved in six cases (54.5%) in the P–M area, nine cases (81.8%) in the C–A area, eight cases (72.7%) in the L–S area, and seven cases (70.0%) in the total area (Table 6). The mean KSPD scores at the infant corrected age of 1.5 years by dose are shown in Table 7. The mean score in the P–M area was 82, in the C–A area was 90, and in the L–S area was 88. In the total area, the mean score of KSPD was 87.

## 4. Discussion

The present study showed that the developmental prognosis of children who had received tadalafil therapy in utero was encouraging. In addition, we explored the safety aspect of tadalafil therapy and the potential adverse effects on the development. Considering that tadalafil treatment is a novel therapeutic agent for FGR, it is essential to evaluate its long-term effects on infants. A systematic review reported neither any apparent severe adverse maternal side effects nor any increases in the rates of stillbirths, neonatal deaths, or congenital anomalies attributed to sildenafil citrate [10]. However, the Dutch STRIDER study reported a significant increase in persistent pulmonary hypertension of neonates [3]. Considering that other studies have not reported such adverse events, the long-term evaluation of the safety of the PDE5 inhibitors, including sildenafil and tadalafil, should be continued.

We adopted the KSPD test for the evaluation of the development of infants in the present study because it is widely used in Japan to evaluate developmental outcomes in high-risk infants (however, its use is limited to Japan). The Bayley Scales of Infant Development (BSID) is a globally well-known and standard representative developmental assessment test. KSPD and BSID have been compared in two previous studies [11,12]. Both reports concluded that there are good correlations between KSPD and BSID, indicating that the motor, cognitive, and language scores, realistically represent specific domains of child development (even though the scales of the two tests are not similar). Therefore, we consider KSPD adequate and acceptable for the evaluation of infant development.

In the present study, we report, for the first time, a developmental evaluation of infants who had received tadalafil in utero as a novel therapeutic treatment for FGR. Although the TADAFER II trial was suspended without completing registration of the original planned number of cases, the study concluded (as a short-term prognosis) that fetal, neonatal, and infant deaths decreased, and that this observation was potentially due to the prolongation of the gestational period in the case of early-onset FGR [1].

The developmental prognosis of children evaluated by KSPD in the present study indicates that infants at a corrected age of 1.5 years made good progress in growth and development. Using the KSPD data from the Neonatal Intensive Care Unit (NICU) network database in Japan as a reference, 20.8% of very low birth weight (VLBW: birth weight is 1000–1500 g) infants achieved a score lower than 70 at the age of three years, and 27.4% achieved a score of 71–84 (unpublished data). In extremely low birth weight infants (birth weight less than 1000 g), 20.8% achieved a KSPD score lower than 70 and 32.7% achieved a score of 71–84. Because the data were associated with low birth weight infants, either with or without FGR, a comparison of the KSPD scores between the present study and the NICU network database is unlikely to be completely accurate. Nonetheless, the KSPD scores achieved in tadalafil-treated infants at a corrected age of 1.5 years were reasonably encouraging (considering KSPD scores in the NICU database were obtained from three-year-olds). If tadalafil therapy shows good developmental prognosis, in addition to the positive short-term outcomes reported in the TADAFER II study (decreased child death), it could be effective in the treatment of FGR. The data for control group are lacking in the present study, although it is essential that these are evaluated. We intend to accomplish this task in a future study.

FGR is associated with preterm birth, and for survivors, an increased risk of motor and sensory neurodevelopmental deficits, cognitive and learning impairments, and cerebral palsy [13,14,15,16,17,18,19,20]. It has been reported that even small-for-gestation age (SGA) children, which is an evaluation based on birth weight, have more cognitive challenges, inattention-hyperactivity symptoms, and school challenges than appropriate-for-gestational age children [19]. von Beckerath et al. discriminate between SGA and intrauterine growth restriction (IUGR) owing to placental insufficiency, and have reported that IUGR has greater cognitive impairments than SGA [20]. In the present study, neurological development was generally without complications, and no cases developed cerebral palsy. Although our population is different from that in the reports above, it is certainly a high-risk population, and careful follow-up is necessary. Each outcome requires further long-term evaluation, such as at school-age.

The KSPD score of one case administered a 40-mg dose of tadalafil was low (P–M: 60; C–A: 81; L–S: 58; TOTAL: 74) and this decreased the overall KSPD score of the 40-mg dose group. Notably, the infant in question may have a developmental disorder and a careful follow-up procedure is planned. Multiple follow-up studies of the development of FGR infants into school-age childhood have revealed deficits in gross and fine motor skills, cognition, memory and academic ability, as well as neuropsychological dysfunctions encompassing poor attention, hyperactivity, and altered mood [21,22,23,24,25]. Hence, additional case studies and a developmental evaluation up to the school age are warranted.

## 5. Conclusions

The growth and developmental progress of infants born of tadalafil-treated mothers at a corrected age of 1.5 years have been found to be satisfactory. A limitation of the study is the lack of a control group and there are no previous reports on the developmental evaluation of children who were FGR in utero based on KSPD; therefore, it is difficult to assess the results of the present study accurately. Consequently, further studies are warranted for additional cases and for evaluating long-term prognosis.

## Figures and Tables

**Figure 1 jcm-09-01448-f001:**
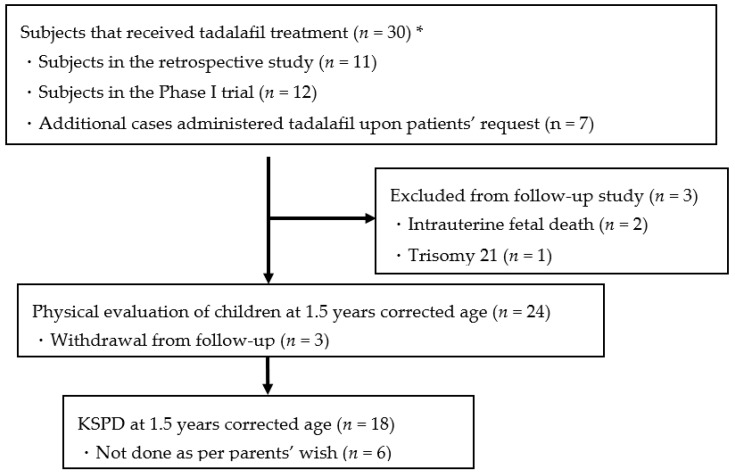
Study profile. * All subjects received tadalafil treatment at Mie University Hospital and their children were scheduled for follow-ups at Mie University.

**Figure 2 jcm-09-01448-f002:**
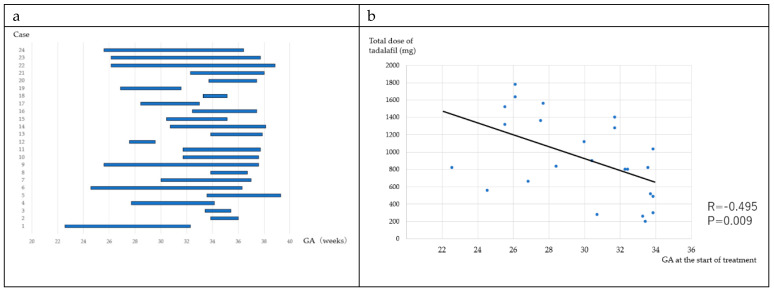
The duration of treatment for each of the patients (**a**) and the correlation between the GA at the start of treatment and total drug dose (**b**). (**a**) The duration of treatment for each patient (**b**) the correlation between the GA at the start of treatment and total drug dose.

**Table 1 jcm-09-01448-t001:** Number of cases treated with tadalafil and number of follow-up cases after birth.

	Tadalafil Dose			
	10 mg	20 mg	40 mg	Total
Total number of cases administered tadalafil	3	20	7	30
Intrauterine fetal death		1	1	2
Trisomy 21		1		
Withdrawal from follow-up		2	1	3
Number of follow-up cases (at least 1.5 years corrected age)	3	16	5	24
KSPD at 1.5 years corrected age)	2	11	5	18

**Table 2 jcm-09-01448-t002:** Characteristics of the cases.

	N = 24 Median (Minimum–Maximum Value)
Gestational age at delivery (days)	258 (207–275)
Gestational age at delivery (weeks)	36.9 (29.6–39.3)
Birth weight (g)	1912 (874–2646)
Birth weight (SD)	−2.0 (−3.3 to 1.6)
Z score of birth weight	−0.54 (−0.82 to −0.23)
Height (cm)	43 (33.0–48.5)
Z score of height	−0.44 (−0.75 to −0.02)
Head circumference (cm)	30.3 (26.5–35.0)
Z score of head circumference	−0.27 (−0.69 to 0.36)
Estimated fetal weight at the start of treatment (g)	1130 (309–1714)
Z score of estimated fetal weight at the start of treatment	−0.41 (−0.68 to −0.28)
Estimated fetal weight at the start of treatment (SD)	−2.1 (−2.8 to −1.5)
Gestational age at the start of treatment (days)	214 (158–237)
Gestational age at the start of treatment (weeks)	30.0 (22.0–33.0)
	SD: standard deviation

**Table 3 jcm-09-01448-t003:** Number of cases, which had catch up within the 3rd percentile.

	10 mg/day (*n* = 3)	20 mg/day (*n* = 16)	40 mg/day (*n* = 5)	Total
BW				
Catch up + (*n*)	2	15	4	21
Catch up − (*n*)	1	1	1	3
Catch up (%)	66.7	93.8	80.0	87.5
Height				
Catch up + (*n*)	2	14	4	20
Catch up − (*n*)	1	2	1	4
Catch up (%)	66.7	87.5	80.0	83.3
Head circumference				
Catch up + (*n*)	3	16	5	24
Catch up − (*n*)	0	0	0	0
Catch up (%)	100	100	100	100

BW: body weight.

**Table 4 jcm-09-01448-t004:** The mean months infant took to catch up within the 3rd percentile.

	10 mg/day (*n* = 3)	20 mg/day (*n* = 16)	40 mg/day (*n* = 5)	Total
BW (months)	9 (4–10)	5.2 (0–12)	7.3 (4–12)	5.8 (0–12)
Height (months)	2 (0–4)	3.8 (0–18)	5.3 (0–12)	3.9 (0–18)
Head circumference (months)	6.7 (0–10)	2.3 (0–10)	7.2 (0–12)	3.8 (0–12)

BW: body weight

**Table 5 jcm-09-01448-t005:** The number of cases and ratio in each DQ area based on KSPD score at 1.5-year infant corrected age.

**DQ**	**P-M**	**C-A**	**L-S**	**TOTAL *n* = 18**
10 mg (*n* = 2)				
≦70	1 (50.0)	0 (0)	0 (0)	0 (0)
70–85	0 (0)	0 (0)	0 (0)	0 (0)
85≦	1 (50.0)	2 (100)	2 (100)	2 (100)
20 mg (*n* = 11)				
≦70	1 (9.0)	0 (0)	2 (18.2)	0 (0)
70–85	4 (36.4)	2 (18.2)	1 (9.0)	3 (27.2)
85≦	6 (54.5)	9 (81.8)	8 (72.7)	8 (72.7)
40 mg (*n* = 5)				
≦70	2 (40.0)	1 (20.0)	2 (40.0)	1 (20.0)
70–85	1 (20.0)	2 (40.0)	0 (0)	2 (40.0)
85≦	2 (40.0)	2 (40.0)	3 (60.0)	2 (40.0)

Data are presented as *n* (%). P-M: Postural-Motor; C-A: Cognitive-Adaptive; L-S: Language-Social.

**Table 6 jcm-09-01448-t006:** DQ score by KSPD for each dose.

*n* = 18				
DQ	P-M	C-A	L-S	TOTAL
≦70	4 (22.2)	1 (5.6)	4 (22.2)	1 (5.6)
70–85	5 (27.8)	4 (22.2)	1 (5.6)	5 (27.7)
85≦	9 (50.0)	13 (72.2)	13 (72.2)	12 (61.7)

Data are presented as *n* (%). P-M: Postural-Motor; C-A: Cognitive-Adaptive; L-S: Language-Social; TOTAL: Total area.

**Table 7 jcm-09-01448-t007:** The mean KSPD score at 1.5-year infant corrected age based on dose.

DQ	10 mg/day (*n* = 2)	20 mg/day (*n* = 11)	40 mg/day (*n* = 5)	TOTAL
P-M	81	85	75	82
C-A	99	91	85	90
L-S	89	91	83	88
TOTAL	94	88	83	87

P-M: Postural-Motor; C-A: Cognitive-Adaptive; L-S: Language-Social; TOTAL: Total area.

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
