# Peer review of "Developmental Evaluation of Infants Who Have Received Tadalafil in Utero for Fetal Growth Restriction"

_jcm, 2020, doi:10.3390/jcm9051448_

Round 1

Reviewer 1 Report

In this study, the authors have assessed the developmental progress of infants whose mothers were administered the PDE-5 inhibitor tadalafil as a potential therapeutic agent for FGR.  However, this reviewer does not see how the study is able to fulfil its aim ‘to assess the long-term effects of tadalafil’ given the complete absence of a control group, who were either administered a placebo or who at the very least would constitute an observational cohort, assessed in the same way as the subjects of the present study following FGR, and at equivalent developmental stages.

In addition to this limitation, which has been recognised by the Authors in the Discussion section but could be considered as a serious limitation significantly reducing the validity of the data presented, there are also several issues within the rest of the manuscript that need to be addressed.

  • Descriptions within the experimental section are lacking. More information needs to be provided on how the infants were selected from the original two studies – did the authors select only those infants whose developmental trajectory was relatively normal compared with others in the FGR/tadalafil-treated cohorts who might have been more severely affected and therefore whose parents would not have been able to participate in this follow-up study?
  • How were the data handled? How were outcome measures assessed for normality; why are mean/SD and median range presented for the characteristics outlined in Table 2?
  • The authors need to provide an explanation and rationale for how and why the data have been age-corrected.
  • Data presented in Tables 5 and 6 appear identical – why have both been included?
  • Why was catch-up growth defined as infants who achieved sizes > 3rd centile, and why are these criteria different to that used to define FGR (i.e. <1.5 SD of EFBW for GA)?
  • Information on how the developmental outcomes are deemed ‘normal’ or not when the comparator group identified in the Discussion section was a) assessed at 3 years of age and b) from a different population of small infants.

Reviewer 2 Report

  • Line 38: The safety profile was confirmed. What was the safety profile? Were any side effects or potential adverse effects for the neonate observed?
  • Line 79: It looks like the children from only one of the participating centres particpated in the long-term follow-up. It would be worthwile to write down or show in a flowchart how many patients in total were treated within the two studies, how many particpated in the current centre and how many were seen for follow-up. Why did this centre perform follow-up, while the other centres did not? It might also be good to describe whether the baseline characteristics of these patients differed from the patients in the other particpating centres to make sure that a representative sample of the original patient group has been evaluated in the current follow-up. 
  • Line 85: Why was the KSPD test not performed in 6 children that were seen at the age of 1,5 years?
  • Table 5 and 6: The term ‘dose’ does not seem to be in the right place in the tables.
  • Table 5 and 6: In the group children treated by 20 mg/day, one child seems to be missing in the total scores.
  • Line 145: Ref 7-14 are mentioned shortly, and conclude that other studies found neurodevelopmental deficits. It would be worthwhile to discuss the main findings of these studies and whether the study populations were comparable to the current population and what the explanation could be for the different findings in the current study.
  • Line 158: As a reader, I would like to know more details on why the current study design was chosen. Why haven’t the control groups of the original studies not be investigated? It is not clear to which results or population the current results can be compared with. And another limitation is the low number of children. Why were not all children from all participating centres invited for follow-up?

Reviewer 3 Report

  1. The article states that enrolled infants had been included in two prior studies. One study had 11 patients (reference 2), and the other had 14 patients (reference 3). The authors should clarify

that the cohort included patients not previously reported. As stated, it is confusing.

  1. Please clarify the gestational age at delivery.
  2. The definition of fetal growth restriction was an estimated fetal weight <1.5 SD. This corresponds to the 7th However, normal catch-up growth was defined as a weight >3rd percentile. Please clarify why the difference in the definition of abnormal growth (in utero vs. postnatal).
  3. There were two fetal deaths in this cohort (6.6%). Were these deaths already reported in the prior studies?
  4. An average estimated fetal weight or head circumference are difficult to interpret, as they are gestational age-dependent. Z scores are more appropriate, as reported.
  5. Please mention the duration of treatment for each of the patients. This is probably best done with a time bar-graph. Dose and duration could then be used to estimate the total amount of medication prescribed. Correlations could then be explored, to include gestational age and total drug dose.
  6. A prior study from the authors was closed prior to complete enrollment due to reported adverse effects of the use of Phosphodiesterase 5 inhibitors in pregnancy (STRIDER studies). The authors should comment on this. However, a recent systematic review (Dunn et al: Fetal Diagn Ther 2017;41:81–88) did not confirm such findings. Therefore, please include this systematic review in the Discussion to preserve the importance of the results of the current study.
  7. Please consider reviewing the manuscript for syntactical analysis.

Round 2

Reviewer 1 Report

We thank the authors for addressing most of my comments from the previously submitted version of this paper.  There are just a few further comments/suggestions below.

1) Figure 1 is helpful and definitely aids the comprehension of the study for the reader. 

2) Out of interest, how does the physician deal with patient requests for experimental medicines, such as Tadalafil? 

3) Line 64: ‘which is considered good neurologic development’: this phrase does not really make sense within the sentence.  Do the authors mean to say that infants attaining body weights above the 3rd centile tend to have normal neurologic development?  Please consider rephrasing.

4) Lines 65-68: The description of age-adjustment might need to be made clearer still to the wider readership of the journal, as defining by ‘age from the estimated date of confinement’ is not standard amongst many countries.  This is presumably related to the estimated date of delivery?  Perhaps consider rephrasing.  Similarly, the phrase “the deliveries with FGR were almost preterm” – do the authors mean that almost all deliveries were preterm?  Again, this needs to be made more clear/accurate.

5) Further proof-reading is still required prior to publication; several issues with grammar throughout (e.g. line 145: ‘developmental prognosis on children’ should be ‘of children’).

Author Response

Response to Reviewer 1 Comments

  1. 1. Out of interest, how does the physician deal with patient requests for experimental medicines, such as Tadalafil?

Our response: On patients request for experimental medicine, such as tadalafil, the administration is deliberated upon in the Medical Quality Committee composed of members, such as doctors and pharmacists, and if approved, the medicine is administered according to the rules of the hospital. When an experimental medicine—not just tadalafil, but any experimental medicine—is administered upon the request of a patient, it is done so only after an extensive deliberation.

  1. 2. Line 64: ‘which is considered good neurologic development’: this phrase does not really make sense within the sentence.  Do the authors mean to say that infants attaining body weights above the 3rdcentile tend to have normal neurologic development?  Please consider rephrasing.

Our response: Thank you for your comment. As you have correctly mentioned, I intended to set the criteria for catch-up that would lead to a good neurological outcome. References [6-8] have been cited in this regard as supporting documents. The sentence has been rephrased to read as follows:

Line 68; “Infants were considered to have caught-up with age-matched normal infants when the values were above the 3rd percentile, which is thought to lead to good neurological development [6-8].

  1. 3. Lines 65-68: The description of age-adjustment might need to be made clearer still to the wider readership of the journal, as defining by ‘age from the estimated date of confinement’ is not standard amongst many countries.  This is presumably related to the estimated date of delivery?  Perhaps consider rephrasing.  Similarly, the phrase “the deliveries with FGR were almost preterm” – do the authors mean that almost all deliveries were preterm?  Again, this needs to be made more clear/accurate.

Our response: Thank you for your comment. The corrected age “corrects” for the prematurity of a baby. It is calculated by subtracting the number of months of prematurity from the chronological (actual) age. The formula is as follows:

[Corrected age] = [Chronological (Actual) age] - [Months preterm]

For example, the baby was born at a gestational age of 32 weeks, the baby was 8 weeks premature (a term pregnancy is 40 weeks; simply subtract baby’s gestational age from 40 weeks to find the number of weeks of prematurity). If the baby’s actual age is 6 months, then the corrected age would be:

6 months – 2 months = 4 months (corrected age)

It is important to correct for the prematurity of a baby to get an accurate assessment of the developmental abilities. Many professionals working with premature infants use corrected age until the infant reaches an age of 2 years. As you have pointed out, we wanted to convey that many deliveries in this study’s population were preterm. Therefore, we adopted the corrected age to evaluate the development of infants.

We have added the following text, and have rephrased the text as follows:

Line 72; “The corrected age was adopted for the developmental evaluation of infants, considering that many deliveries in the population included in this study were preterm. It enhances the reliability of evaluations. The corrected age was calculated by subtracting the number of months of prematurity from the chronological (actual) age.

  1. Further proof-reading is still required prior to publication; several issues with grammar throughout  (e.g. line 145: ‘developmental prognosis on children’ should be ‘of children’).
    Our response: Thank you for your suggestion. We have got the manuscript re-edited by Editage, a professional English language editing company.

As suggested by you, we have replaced the word “on” with “of” in the phrase pointed out by you (line 152 in the revised manuscript).

Reviewer 2 Report

  • Line 38: The safety profile was confirmed. What was the safety profile? Were any side effects or potential adverse effects for the neonate observed?
  • Figure 1 is useful, thank you for adding.
  • Why was chosen to not investigate the control groups of the original studies? This is a consideration that needs to be discussed.

Author Response

Response to Reviewer 2 Comments

  1. 1. Line 38: The safety profile was confirmed. What was the safety profile? Were any side effects or potential adverse effects for the neonate observed?

Our response: Although we had verified the safety in phase I trial, in the TADAFER II study, we mainly focused on the safety of tadalafil for mothers and neonate. TADAFER II was suspended based on the recommendation of Japan Agency for Medical Research and Development (AMED) in view of the results of the STRIDER-UK study showing negative effects of sildenafil in the treatment of FGR. Because AMED has also recommended that the TADAFER II study should mainly evaluate the safety, this study was focused on the safety aspect. For this reason, we mentioned that the “safety was confirmed” in this manuscript. In contrast to the occurrence of neonatal persistent pulmonary hypertension caused by sildenafil in the Dutch STRIDER study, no adverse events related to tadalafil were seen in mothers and neonates in the TADAFER II study.

We have added the following text in the revised manuscript (Introduction section):

Line 37; “Although the TADAFER II study was suspended in view of the results of the STRIDER-UK trial showing negative effects of sildenafil in the treatment of FGR [2] and the recommendation of Japan Agency for Medical Research and Development (AMED) based on these results, the study demonstrated a decrease in the mortality rate of FGR fetuses, neonates, and infants.

Line 41; “In addition, contrary to the occurrence of neonatal persistent pulmonary hypertension due to sildenafil in the Dutch STRIDER study [3], no adverse events related to tadalafil were seen in mothers and neonates in the TADAFER II study.

  1. 2. Why was chosen to not investigate the control groups of the original studies? This is a consideration that needs to be discussed.

Our response: Thank you for your comment. Unfortunately, we have not previously planned enough, a strict, long-term follow-up of untreated infants. We did not follow-up the control group. This is the reason why we could not investigate the control group from the original study. The evaluation of the control group should indeed be performed, and it is our objective in a future study.

We have added the following text in the Discussion section:

Line 188; “The data for control group is lacking in the present study, although it should be evaluated essentially. We intend to accomplish this task in a future study.